# Chiral Porphyrin Assemblies Investigated by a Modified Reflectance Anisotropy Spectroscopy Spectrometer

**DOI:** 10.3390/molecules28083471

**Published:** 2023-04-14

**Authors:** Ilaria Tomei, Beatrice Bonanni, Anna Sgarlata, Massimo Fanfoni, Roberto Martini, Ilaria Di Filippo, Gabriele Magna, Manuela Stefanelli, Donato Monti, Roberto Paolesse, Claudio Goletti

**Affiliations:** 1Department of Physics, Università di Roma Tor Vergata, Via della Ricerca Scientifica 1, 00133 Roma, Italy; 2Department of Chemical Science and Technologies, Università di Roma Tor Vergata, Via della Ricerca Scientifica 1, 00133 Rome, Italy; 3Department of Chemistry, Sapienza Università di Roma, Piazzale Aldo Moro 5, 00185 Rome, Italy

**Keywords:** circular dichroism, porphyrins, chirality, chiral layers, supramolecular chirality, reflectance anisotropy spectroscopy

## Abstract

Reflectance anisotropy spectroscopy (RAS) has been largely used to investigate organic compounds: Langmuir–Blodgett and Langmuir–Schaeffer layers, the organic molecular beam epitaxy growth in situ and in real time, thin and ultrathin organic films exposed to volatiles, in ultra-high vacuum (UHV), in controlled atmosphere and even in liquid. In all these cases, porphyrins and porphyrin-related compounds have often been used, taking advantage of the peculiar characteristics of RAS with respect to other techniques. The technical modification of a RAS spectrometer (CD-RAS: circular dichroism RAS) allows us to investigate the circular dichroism of samples instead of the normally studied linear dichroism: CD-RAS measures (in transmission mode) the anisotropy of the optical properties of a sample under right and left circularly polarized light. Although commercial spectrometers exist to measure the circular dichroism of substances, the “open structure” of this new spectrometer and its higher flexibility in design makes it possible to couple it with UHV systems or other experimental configurations. The importance of chirality in the development of organic materials (from solutions to the solid state, as thin layers deposited—in liquid or in vacuum—on transparent substrates) could open interesting possibilities to a development in the investigation of the chirality of organic and biological layers. In this manuscript, after the detailed explanation of the CD-RAS technique, some calibration tests with chiral porphyrin assemblies in solution or deposited in solid film are reported to demonstrate the quality of the results, comparing curves obtained with CD-RAS and a commercial spectrometer.

## 1. Introduction

Reflectance anisotropy spectroscopy (RAS) is a surface-sensitive optical technique with significant peculiarities: it strongly reduces or avoids the contamination or even damage of the sample (when electrons or charged particles are used as probes in other experimental techniques), is utilizable in vacuum, in atmosphere or in transparent media (also in liquids) without limitations due to pressure, allows for the investigation of insulating samples without problems of charging, and gives the possibility to study the structure of surfaces/layers and even buried or immersed interfaces [1,2,3].

RAS has been applied to clean surfaces of metals and semiconductors in ultra-high vacuum (UHV), boosting a significant development of the knowledge in surface science [4,5,6,7], then to low-dimensionality solid state systems (films, wires, dots) [8,9], and finally to organic samples (for example, samples grown by organic molecular beam epitaxy (OMBE) [10,11] or porphyrin and sapphyrin layers deposited by Langmuir–Blodgett and Langmuir–Schaeffer technique) [12,13]. Clean surfaces of metals and semiconductors immersed in solutions and organic and biological layers deposited in liquid have been recently studied, down to the characterization of samples in electrolytes, monitoring, for example, in real time and in situ by RAS and scanning tunneling microscope (STM), the electrochemical reactions at the Cu(110)–liquid interface triggered by voltage cycles applied to the sample [14,15,16].

In all these experiments, RAS has been limited to the anisotropy of the linear dichroism of matter. As we will show hereinafter, the same technique can be used to investigate the circular dichroism in transmittance by a proper modification in the experimental apparatus, opening intriguing perspectives in the experimental study of chirality. This topic constitutes one of the most active and dynamic research areas related to varied disciplines such as chemistry, physics, biology, and material science [17].

Chirality is a characteristic that pervades the universe, from the small range of the subatomic particles to the immensity of the spiral galaxies: it is defined as the geometric property of a rigid object (or spatial arrangement of points or atoms) of being non-superposable on its mirror image [18]. This term derives from the Ancient Greek word “cheir” (χείρ) for hand, to give a pictorial sketch of the chirality meaning.

In chemistry, molecules that feature chirality can be spatially arranged into two specular, nonsuperimposable structures called enantiomers [19]. Since enantiomeric pairs are the same chemical species, the different spatial arrangement does not induce physical or chemical changes in the properties of these two isomers unless they are placed or interact with an asymmetrical environment. To have an idea of the importance of chirality, it is sufficient to note that all living systems are inherently chiral and represent an asymmetrical environment since, during evolution, only a single enantiomer has been selected for the synthons of essential biological macromolecules, such as L-aminoacids for proteins and D-sugars for nucleic acids. As a consequence, chirality plays a fundamental role in biological processes, driving the selectivity of most interactions essential for life.

Although conventionally considered important at the molecular level, especially in the pharmaceutical field, chirality is even more pursued at the supramolecular level to produce novel artificial chiral materials suitable for emerging applications such as circular polarizers, chiral chromatography, and (bio)sensor devices. Among the countless combinations of assembling methods and building blocks chosen, porphyrins have been intensively investigated since they offer the possibility to easily modulate molecular interactions, leading to stereospecific chiral systems by means of different approaches that use both chiral and achiral porphyrin units [20]. Additionally, the high absorptions in the ultraviolet (UV) and visible range possessed by these chromophores guarantee strong exciton coupling over large distances that result in diagnostic chiroptical signals unveiling the specific molecular orientation of the macrocycles within the aggregated species [21,22].

We are traditionally interested in characterizing the aggregation mechanisms of chiral metalloporphyrins with an appended proline group in hydroalcoholic solutions [23]. More recently, we have broadened this interest in the preparation of chemical sensors for chiral discrimination by using materials based on the same chiral porphyrins [24]. The enantiorecognition with chemical sensors is particularly challenging since these devices can rely only on a single binding event for chiral recognition. Additionally, controlling the deposition of chiral layers onto the surface of transducers constitutes another critical step since additional factors that may affect the film morphology/chiroptical properties, such as solvent, concentration, and evaporation time should be considered during the formation of sensing films [25].

Here, we investigated the possibility of utilizing a RAS spectrometer in transmission to measure features of porphyrins in solution and in solid films related to circular dichroism (CD), comparing the spectra obtained with the ones produced by commercial extended wavelength CD (ECD) apparatuses [26]. The possibility of using a RAS spectrometer with this new configuration for investigating CD (hereafter indicated as circular dichroism RAS, CD-RAS) could in principle represent just another way of performing experiments that are possible with the existing commercial instruments and then of questionable utility. However, unlike commercial CD spectrometers, the “open structure” of our CD-RAS spectrometer allows for higher flexibility in designing the optical path, making it possible to couple optics with UHV systems, with electrochemical cells, or more generally with other experimental configurations that otherwise should be impossible to configure. The increasing importance of experiments investigating the time domain of the behavior of chiral films (from evolution of the chiral signal after deposition to the investigation of kinetics when exposed to chiral molecules) suggests to go beyond the quite rigid methodology of application of the optical methods normally available in laboratory.

## 2. Experimental Methods and Techniques

The RAS signal is defined as the ratio ΔR/R between the difference ΔR of the light intensity R_α_ and R_β_ reflected by the sample for a beam polarized into two different and independent polarization states α and β, and their average R = (R_α_ + R_β_)/2, as a function of photon energy [2,3]:(1)$$\frac{\Delta R}{R}=2\frac{R_{\alpha }-R_{\beta }}{R_{\alpha }{+R}_{\beta }}$$

In the case of linearly polarized light, the electric field is directed along two orthogonal directions α and β, usually aligned with the main anisotropy axes of the sample surface plane. In this version, a RAS spectrometer is essentially an ellipsometer at near normal incidence, whose experimental set-up strongly simplifies the interpretation of data with respect to traditional ellipsometry. A (sometimes complex) deconvolution from rough data is still necessary to correctly understand and explain the obtained results by modeling [2,3]: but the existence of a non-vanishing anisotropy often represents a meaningful result per se, suitable to characterize a certain physical system, or to follow its dependence upon the variation of definite experimental conditions as temperature, contamination, strain, coverage, pressure, electric field, etc., finally defined as the signature of the existence of a certain, well-defined phase at the surface. Linearly polarized RAS has been widely applied to investigate surfaces and interfaces, more generally 2D systems, exploiting the different symmetries of the bulk/substrate with respect to the surface/top layer [4,5,6,7,15]: the signal anisotropy coming from centrosymmetric crystals or amorphous substrates is ideally null, and then, any anisotropy signal measured by RAS is recognized and isolated as due to the surface/top layer [2,3]. In the experiments reported in Figure 1, α and β were parallel to the [$\bar{2}$11] and [1$\bar{1}$0] directions on the (111) cleavage plane of a diamond sample 2 × 1 reconstructed. The evident peak dominating the spectrum is due to optical transitions between electronic surface states of the π-bonded reconstructed surface [27].

In a typical RAS system, light from a source (usually Xe or tungsten lamp, emitting photons in the near UV–visible–near infrared (IR) range, that is 280–1000 nm) is shined and focused into a polarizer, then onto a photoelastic modulator (PEM) and finally onto the sample (properly oriented), eventually passing through a special low-birefringence window if the sample is in ultra-high vacuum (UHV) or in liquid. The PEM (properly driven by an oscillating circuit at the resonance frequency ν_0_ of the piezoelectric crystal) introduces a phase shift equal to ±π between light beams propagating along ordinary and extraordinary axes, thus modulating the linear polarization of light between two orthogonal, independent states *x* and *y*. The light intensity reflected by the sample for two polarizations (R*_x_* and R*_y_*) is then collected and focused onto a second polarizer (analyzer) and finally into a monochromator. At the exit slit, there is a detector (photomultiplier, photodiode, etc.) chained to a preamplifier and then to a lock-in amplifier to filter the signal, tuned at the correct modulation frequency (exactly 2ν_0_, that is the second harmonic of the modulated signal). Another version of a RAS spectrometer exists, where the analyzer is not present.

If the PEM introduces a phase shift equal to ±π/2, the outgoing light is alternatively polarized between right and left circularly polarized light, with a resulting signal carrying information about the circular dichroism of the shined sample. The high modulation frequency (about 50 kHz) favors the signal stability, eliminating the mechanical noise and the low frequency modes. In Figure 2, we draw a time-sketch of the successive polarization states of circularly polarized light after the passage through the PEM: the oscillation frequency between circularly left and circularly right polarization states is exactly coincident with the oscillation frequency of the PEM. In the signal, a linear polarization contribution also appears (just for one of the two orthogonal independent states, we will call it “α”), at a frequency that is twice the PEM frequency. Just tuning the lock-in at a frequency equal to ν_0_, the only contribution due to the circularly polarized light is selected.

The technical equipment for the CD-RAS experiments is sketched in Figure 3. After being transmitted through the sample, the beam is reflected on an oxidized Si(100), perfectly isotropic for linearly polarized light, not introducing any artifacts in the signal. Strictly speaking, the term “reflectance” in the acronym CD-RAS is inappropriate here, as the experiments are conducted in transmission: reflectance anisotropy for circularly polarized light is rigorously null. This can be demonstrated using the formalism of the Jones matrices [28]; however, an immediate and identical conclusion is reached by evaluating the effect on the beam of the surface/layer (that is crossed twice). We have experimentally verified this null result for chiral films whose thicknesses were down to 10 nm.

The apparatus is highly efficient (as we will show in the next section) in extracting the CD-related features in transmission from solutions and thin solid-state layers deposited onto transparent substrates. For CD-RAS, more properly and more generally, we will then define the signal as: (2)$$\frac{\Delta I}{I}=2\frac{I_{\alpha }-I_{\beta }}{I_{\alpha }{+I}_{\beta }}$$
where I_α_ and I_β_ are the light intensities (in this case, transmitted) measured after passing through the sample for a beam alternatively polarized into two different and independent circular polarization states α and β. The “atout” of our CD-RAS (with respect to more traditional apparatuses commonly available, such as the JASCO CD spectrometer [26]) is a more flexible design of the optical path that can be tuned (in length), with a careful setting of the whole experimental setup, according to the peculiar necessity of the experiments, to be coupled with UHV systems, electrochemical cells, or transparent recipients larger that the cuvette used in commercial CD spectrometers. 

The data obtained by CD-RAS were compared with the ones obtained by a commercial spectrometer [26]. While CD-RAS data represent the intensity modulation ∆I/I of the signal when passing from the right circularly polarized state to the left circularly polarized state (see Formula (2)), a commercial spectrometer measures the ellipticity due to the sample, which is a composition (generally associated with the elliptic polarization of light) of the two circularly polarized states, with the same intensity when radiation impinges on the sample, but is then differently absorbed. CD spectra are reported as ellipticity, θ, and are measured in units of mdeg. The ellipticity [θ] is defined as:[θ] = A (*ε*_L_ − *ε*_R_) c *l*
(3)
where *ε*_L_ and *ε*_R_ are the molar extinction coefficients for light left- and right-handed polarized ones, respectively; *l* is the optical pathway of the sample, and c is its concentration. A is a constant with appropriate dimensions.

It is possible to demonstrate that the ellipticity [θ] and the ∆I/I result of a CD-RAS experiment are proportional [26]:[θ] ~ ΔI/I(4)

## 3. Results and Discussion

In this section, the CD-RAS spectra measured in transmission mode for different samples are presented and compared with the spectra recorded by a commercial spectrometer for circular dichroism: i.A solution of chiral porphyrins with a chiral signature (in terms ellipticity) in the order of some thousands of millideg (samples A1 and A2);ii.A solution of chiral porphyrins with a chiral signal in the order of some millideg (sample B);iii.A thin chiral film of porphyrins deposited onto glass substrates, in two enantiomeric configurations (samples C1 and C2).

In cases (i) and (ii), the spectrometer used in CD-RAS experiments was in the Aspnes version, with two polarizers [30]. In case (iii), we used the Safarov–Berkovits version of the CD-RAS spectrometer (without the analyzer), sketched in Figure 3 [29]. In all the experiments, the beam after transmission through the sample was reflected on an oxidized Si(100), perfectly isotropic for linearly polarized light, and then in principle not introducing artifacts in the signal. The spectra were measured in the range 380 nm < λ < 500 nm (where the main molecular contribution was expected in all cases), although the covered possible range is more extended (300 nm < λ < 700 nm).

The CD-RAS spectrometer was first tested on chiral porphyrin aggregates characterized by intense CD-related signals in solution (samples A1 and A2). In Figure 4, the CD-RAS spectrum (red curve) and the JASCO spectrum (black curve) of Sample A1 are reported and compared. The different axes (with different units) are also drawn. The two curves are in excellent agreement. A similar result was obtained with sample A2 (Figure 5). In this case, with a different molecule and different CD activity, the curves exhibit an excellent overall similitude, with the same lineshape at a very good accuracy. The CD-RAS spectrum was here corrected by subtracting the background measured (in the same experimental conditions) with an achiral solution.

The system was then tested on sample B to check the performance with a sample exhibiting three orders of magnitude lower CD-related signals in the millideg range. Given this low value of the CD, the CD-RAS spectrum reported in Figure 6 was obtained by a longer integration time (2 s/spectral point) to gain a higher signal-to-noise ratio, and from the resulting curve, the optical background of the glass cuvette containing the solution (being a quite larger signal due to the birefringence of the vessel) was subtracted. The good agreement of the two curves in this case allows us to demonstrate the sensitivity of CD-RAS at a higher sensitivity scale.

Finally, chiral porphyrin films (samples C1 and C2) deposited onto a glass substrate were measured in transmission mode. The films resulting from letting one 50 μL droplet dry out were approximately 10 nm thick (estimated from atomic force microscope images). In Figure 7, the CD-RAS spectra (red and black curves in panel A) and the JASCO spectra (red and black curves in panel B) are shown for both enantiomers, deposited separately following the same protocol. The background spectrum measured in the same configuration for a clean glass substrate identical to the one used for deposition was always subtracted from the CD-RAS data, to eliminate the spurious anisotropy signal due to the experimental configuration. The final agreement is fully satisfactory, with an excellent signal-to-noise ratio in the CD-RAS curve (integration time: 4 s/spectral point). The broader structures in panel A are due to the larger bandpass of the monochromator used in the CD-RAS spectrometer.

## 4. Materials and Methods

Details about preparation, kinetics and theoretical explanation of samples A1, A2, and B1 are reported in a recent review about the stereospecific self-assembly processes of porphyrin–proline conjugates [23]. In detail, sample A1 contains II-type aggregates of ZnP(L)Pro(–) in hydroalcoholic solution (EtOH:water 25:75, 5 μM); sample A2 contains aggregates of H_2_P(L)Pro(–) in hydroalcoholic solution (EtOH:water 25:75, 10 μM); sample B contains I-type aggregates of ZnP(D)Pro(–) in hydroalcoholic solution (EtOH:water 25:75, 5 μM). Solid samples were prepared as reported in Ref. [25]. In detail, 50 μL of a 0.1 mM solution in toluene of ZnP(D)Pro(–) or ZnP(L)Pro(–) was cast on a ultraflat glass slide to obtain samples C1 and C2, respectively.

The solutions and casted films were analyzed by a JASCO J-1500 Circular Dichroism Spectrophotometer, equipped with a thermostated cell holder set at 298 K and purged with ultra-pure nitrogen gas. The slit width was set to 2 nm and the scanning speed to 20 mdeg/min in continuous scanning mode. Linear dichroism contribution (LD) was found to be <0.0004 DOD units with respect to the baseline in all the cases examined.

AFM measurements were performed in air using a NanoSurf NaioAFM instrument. Experiments were carried out in tapping mode by using silicon tips with a spring constant of about 48 N/m and a curvature radius of less than 10 nm. The film thickness was estimated as the height differences between the molecule layer and the glass slide, made by excavating across the spot with a needle previously dipped in dichloromethane. The average height of the structures emerging from the glass is approximately 12 ± 3 nm, in a region of 30 × 30 µm^2^.

## 5. Conclusions

In this article, we have shown that destructuring a spectrometer for circularly polarized light (CPL) by adapting a RAS system—normally used for linearly polarized light—to CPL, a flexible apparatus is now available to measure, with high sensitivity, the circular dichroism of solutions and very thin solid state films. As a consequence, a new class of experiments would then become accessible as an investigation in real time of the chiral thin film growth from solution drops deposited onto a transparent substrate (that on a vertical glass would undoubtedly glide toward the bottom, while the CD-RAS system can be mounted in a vertical plane, with the sample horizontal). The development of different approaches for the characterization of chiral layers is an important opportunity: the application of a modified RAS spectrometer for CD measurements could represent a significant boost for research in this field.

## Figures and Tables

**Figure 1 molecules-28-03471-f001:**
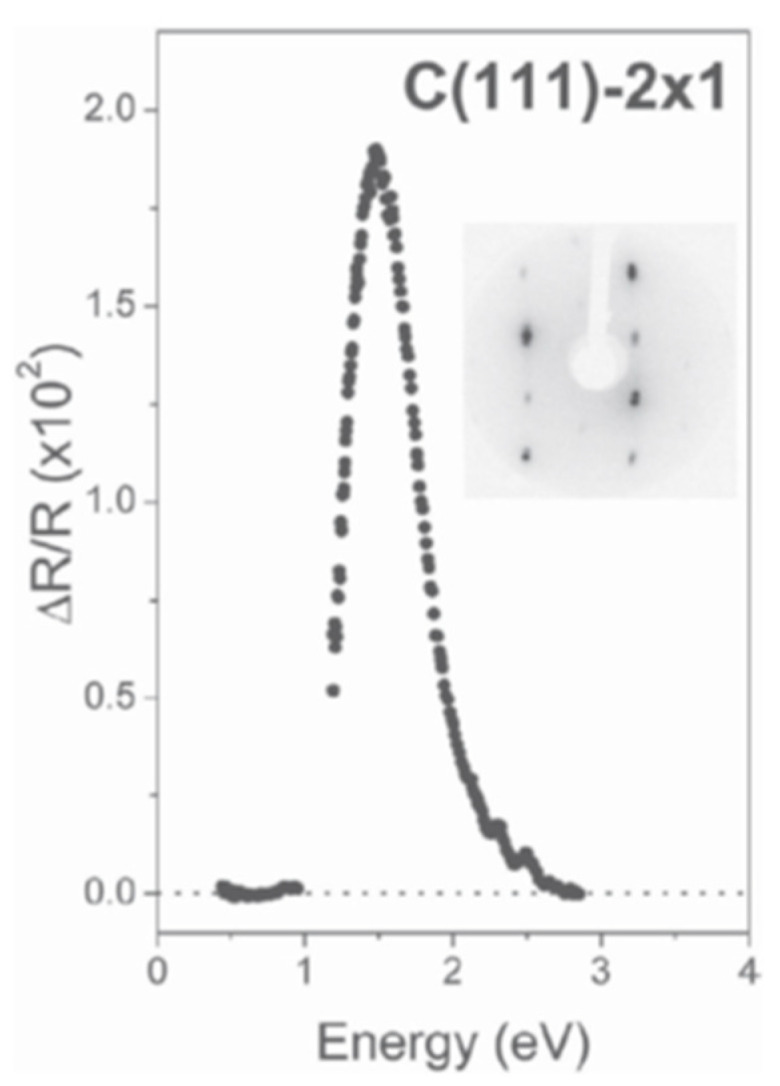
ΔR/R as a function of photon energy for a single domain C(111)-2 × 1 surface, in the energy range from 0.4 to 2.8 eV. The sharp peak (whose sign means light electric field along the 1D chains of the reconstructed surface) is related to optical transitions between the surface electronic bands. The inset shows a low-energy electron diffraction (LEED) picture taken at 70 eV, demonstrating the existence on the reconstructed surface of a largely dominating single domain. Doubling of the spots is due to the regular array of surface steps. More extended experimental details are reported in Ref. [27]. Reproduced from G. Bussetti et al., Europhys. Lett. 2007, 79, 57002, with permission.

**Figure 2 molecules-28-03471-f002:**
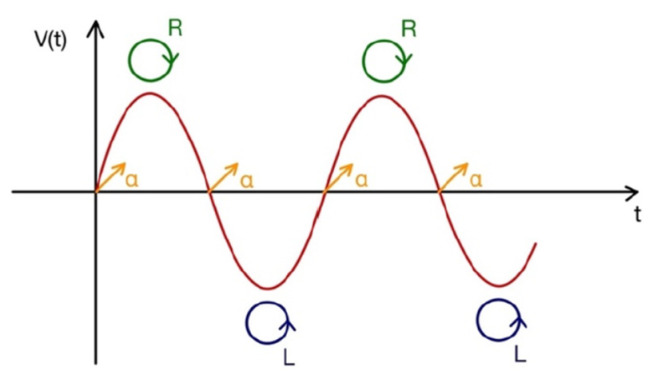
Sketch of the polarization of the light beam after the PEM, for an applied voltage value such that the phase shift is Δ*φ* = π/2. The red curve represents (vs. time) the voltage driving the PEM at the frequency ν_0_. At V = 0 volt, the polarization is linear, here directed as α (yellow arrows). When the voltage reaches its maximum, the resulting polarization is circular and right handed (green circles). When the voltage reaches its minimum, the resulting polarization is circular and left handed (blue circles). The oscillation frequency of the signal between the two states of circular polarization is evidently ν_0_. The frequency of the linearly polarized signal is 2 ν_0_.

**Figure 3 molecules-28-03471-f003:**
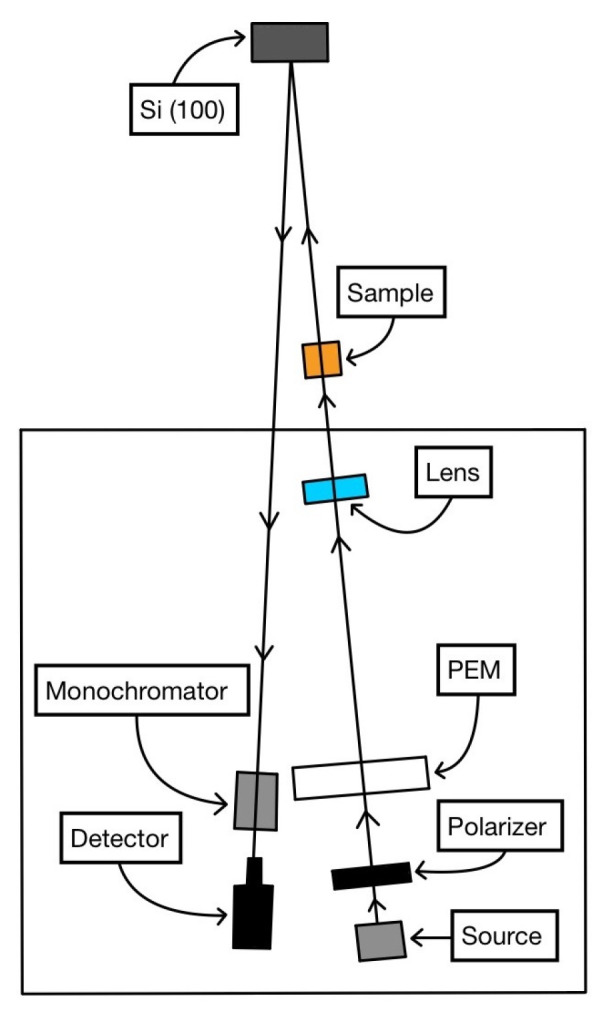
Sketch of the CD-RAS experimental spectrometer. The version here reported is the Safarov/Berkovits-like (see Refs. [1,2,3,29]), with one polarizer. After passing the sample, the light beam is reflected back by an isotropic Si(100) sample. Although the system was mounted horizontally in this case, it can also work vertically.

**Figure 4 molecules-28-03471-f004:**
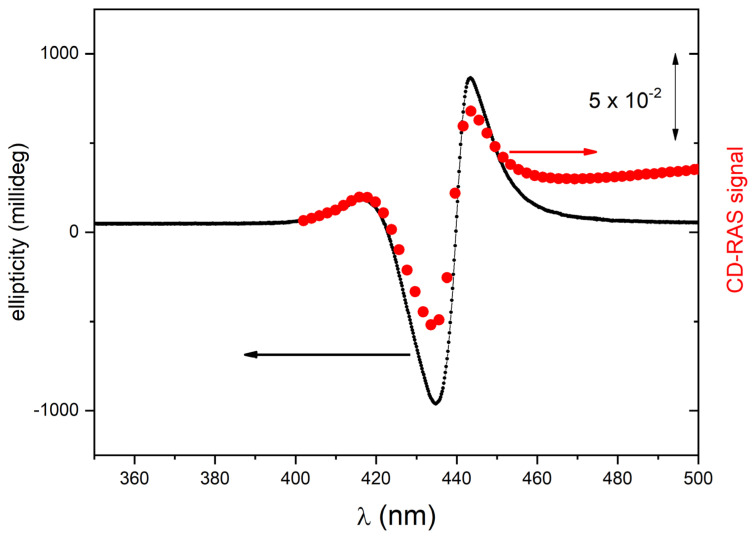
CD-RAS experimental spectrum for sample A1 (red curve, right vertical axis), compared with the ellipticity spectrum measured for the same sample by a commercial spectrometer for circular dichroism (black curve, left vertical axis). The CD-RAS spectrum is reported as measured, without correction. The double-arrow reported in the figure is the unit for the CD-RAS data. Here, sample A1 is a solution containing II-type aggregates of ZnP(L)Pro(–) in hydroalcoholic solution (EtOH:water 25:75, 5 μM).

**Figure 5 molecules-28-03471-f005:**
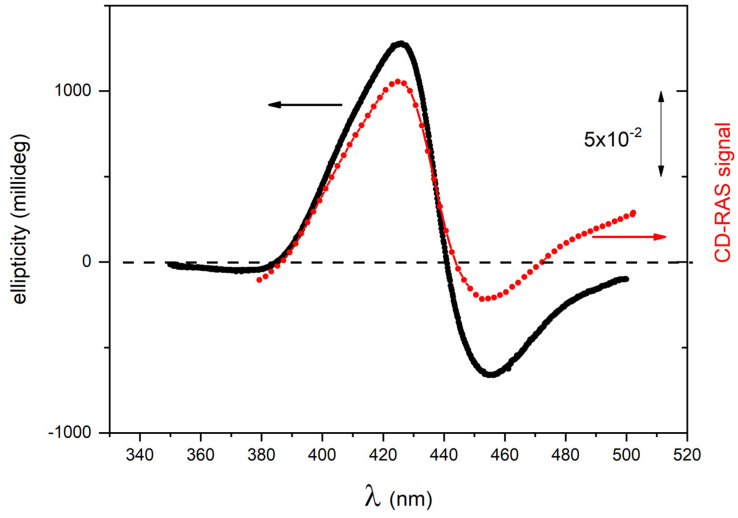
CD-RAS experimental spectrum for sample A2 (red curve, right vertical axis), compared with the ellipticity spectrum measured for the same sample by a commercial spectrometer for circular dichroism (black curve, left vertical axis). The double-arrow reported in the figure is the unit for the CD-RAS data. The CD-RAS spectrum was corrected by subtracting the background measured (in the same experimental conditions) with an achiral solution. Here, sample A2 is a solution containing aggregates of H_2_P(L)Pro(–) in hydroalcoholic solution (EtOH:water 25:75, 10 μM).

**Figure 6 molecules-28-03471-f006:**
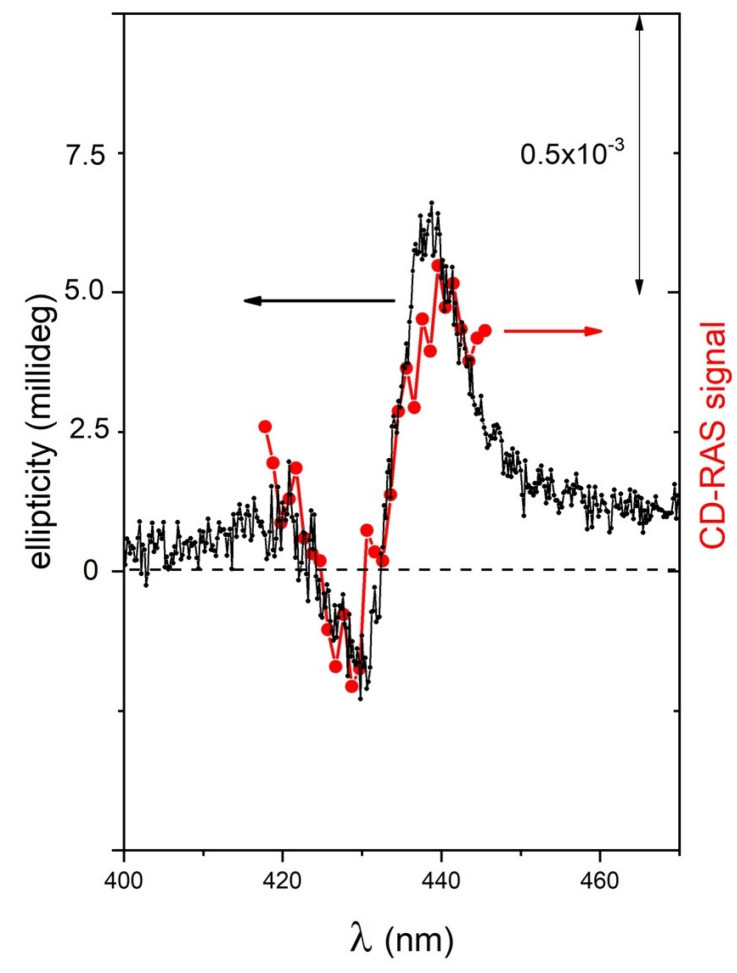
CD-RAS experimental spectrum for sample B (red curve), compared with the ellipticity spectrum measured for the same sample by a commercial spectrometer for circular dichroism (black curve, left vertical axis). The double-arrow reported in the figure is the unit for the CD-RAS data. The CD-RAS spectrum was corrected by subtracting the optical background of the glass cuvette containing an achiral solution (measured in the same experimental conditions). Here, sample B is a solution containing I-type aggregates of ZnP(D)Pro(–) in hydroalcoholic solution (EtOH:water 25:75, 5 μM).

**Figure 7 molecules-28-03471-f007:**
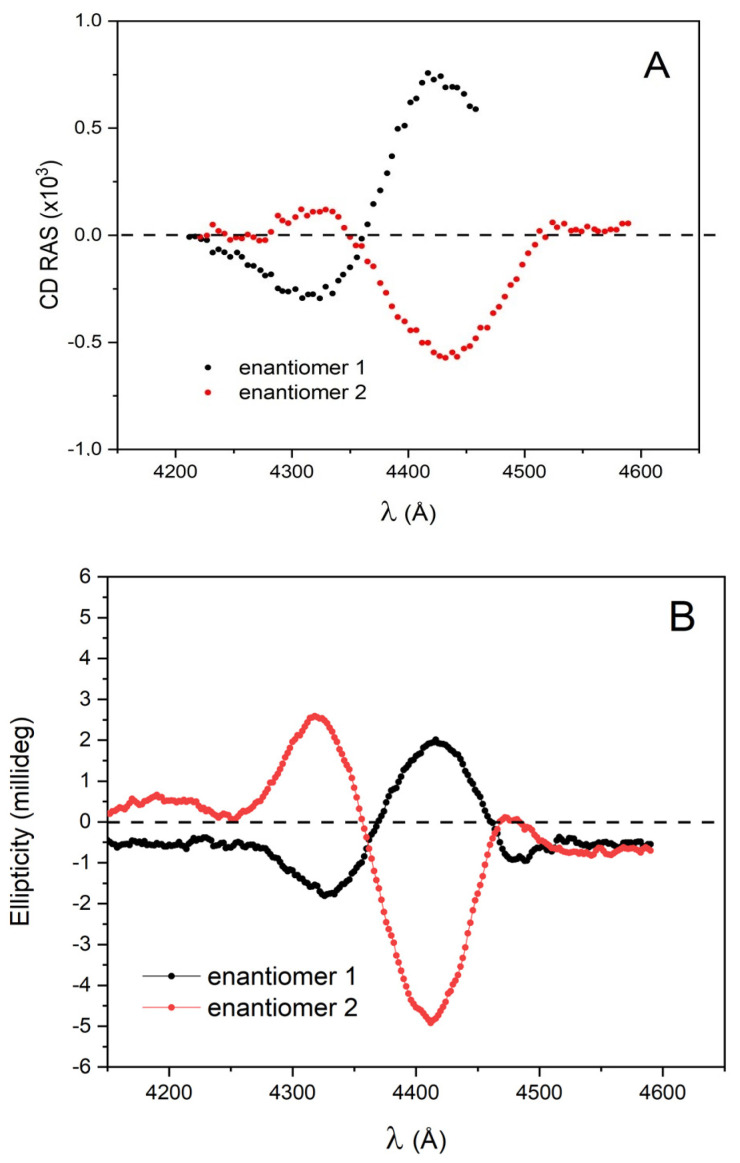
Panel (**A**): CD-RAS experimental spectra for sample C1 (red curve: enantiomer 1) and sample C2 (black curve: enantiomer 2) (see text). The two enantiomers were obtained from a 50 μL droplet of a ZnOEP solution in dichloromethane deposited onto a glass substrate and measured in transmission mode. The resulting film has an average thickness estimated in the range of 10–15 nm from atomic force microscope images (not shown). Panel (**B**): Ellipticity spectra measured on the same samples by a commercial spectrometer for circular dichroism for sample C1 (red curve: enantiomer 1) and sample C2 (black curve: enantiomer 2).

## Data Availability

The data presented in this study and Appendix A are available on request from the corresponding author.

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
