# Peer review of "Chiral Porphyrin Assemblies Investigated by a Modified Reflectance Anisotropy Spectroscopy Spectrometer"

_molecules, 2023, doi:10.3390/molecules28083471_

Round 1

Reviewer 1 Report

The main topic of the paper is introduction of RAS as a way to estimate CD spectrum. It's a interesting, if a bit of a niche application. 

It is important to highlight that this method does not allow for measuring of CD spectra - it only estimates it. The title and apropriate paragraphs in the text need to be modified to reflect this. 

Another issue is that the title suggest that CD can be measured in reflectance mode, which is not true, this is still a transmission measurement, the same as traditional CD, so this should be stated clearly in the abstract and the text. 

As authors highlight the only real advantage of this method is potential application to thin films grown horizontally on transparent substrates. 

The use of word "apparatus" throughout the text to describe an instrument is a bit unusual. 

Reviewer 2 Report

Manuscript ID: molecules-2301478

“Circular dichroism of chiral porphyrin assemblies by a modi-2 fied reflectance anisotropy spectroscopy apparatus.”

By: I. Tomei, B. Bonanni, A. Sgarlata, M. Fanfoni, R. Martini, I. Di Filippo, G. Magna, M. Stefanelli, D. Monti, R. Paolesse and C. Goletti

This manuscript describes the achievements from two groups, one composed of physicists and one of physico-organic chemists in setting up an apparatus for the measurement of CD (let us call it that way, even though it might be inappropriate in certain instances) for complicated samples, especially in the form of films at low temperature. The spectroscopic technique is RAS-CD and due comparison with standard CD is provided; the samples are porphyrin-proline conjugates in various forms, with or without metals, in solution or as films. I am certainly in favor of publication of this paper, subject to the following minor points:

1)In Figures 4 and 5 CD spectra of the order of thousands mdeg are reported. This is what I meant above as supposed (pseudo?) CD. Of course in the literature several examples of that sort are present and I am not saying at all that the recorded signals are fakes. What I am asking is: can the author provide the corresponding UV spectra (from the Jasco apparatus e.g.)? This could help the reader decide (or get an idea) whether the observed signal is due to birefringence (either circular or linear, as it happens with gels, where one observes a broad “absorption” feature) or to something else (distortion from too large absorption). The Authors either could give the data in an added supplementary file or describe it in words in the Materials and Methods section.

2)On the right of Figures 4, 5 and 6 a double-arrow sign is reported, which I surmise being the unit for the RAS-CD experiment. If so, please describe it in the Figures’ captions. Also: is the relation between the mdeg-CD unit and the RAS-CD merely empirical or could the authors rationalize it?

3)The paper is excellently written. However I’d like to suggest the following quite minor changes. Line 120: instead of “a not null”, “a non-vanishing”. Line 202: instead of “when impinge on the sample”, “when radiation impinges on the sample”. Line 255: instead of “a well larger”, “a quite larger” or just “a larger”. Line 273: instead of:  “The films resulting after drying a 50 ul droplet had an average thickness in the order of 10 nm” “The films resulting from letting one 50 ul droplet dry out were approximately 10 nm thick”.
